# Research on Anomaly Network Detection Based on Self-Attention Mechanism

**DOI:** 10.3390/s23115059

**Published:** 2023-05-25

**Authors:** Wanting Hu, Lu Cao, Qunsheng Ruan, Qingfeng Wu

**Affiliations:** University of Xiamen, Xiamen 361005, China

**Keywords:** anomaly detection, feature engineering, attention mechanism

## Abstract

Network traffic anomaly detection is a key step in identifying and preventing network security threats. This study aims to construct a new deep-learning-based traffic anomaly detection model through in-depth research on new feature-engineering methods, significantly improving the efficiency and accuracy of network traffic anomaly detection. The specific research work mainly includes the following two aspects: 1. In order to construct a more comprehensive dataset, this article first starts from the raw data of the classic traffic anomaly detection dataset UNSW-NB15 and combines the feature extraction standards and feature calculation methods of other classic detection datasets to re-extract and design a feature description set for the original traffic data in order to accurately and completely describe the network traffic status. We reconstructed the dataset DNTAD using the feature-processing method designed in this article and conducted evaluation experiments on it. Experiments have shown that by verifying classic machine learning algorithms, such as XGBoost, this method not only does not reduce the training performance of the algorithm but also improves its operational efficiency. 2. This article proposes a detection algorithm model based on LSTM and the recurrent neural network self-attention mechanism for important time-series information contained in the abnormal traffic datasets. With this model, through the memory mechanism of the LSTM, the time dependence of traffic features can be learned. On the basis of LSTM, a self-attention mechanism is introduced, which can weight the features at different positions in the sequence, enabling the model to better learn the direct relationship between traffic features. A series of ablation experiments were also used to demonstrate the effectiveness of each component of the model. The experimental results show that, compared to other comparative models, the model proposed in this article achieves better experimental results on the constructed dataset.

## 1. Introduction

After entering the 21st century, the scale of the internet has grown rapidly, and modern society increasingly relies on network services with more and more terminal devices connected to the internet. This has resulted in an unprecedented volume of network data transmission and an increasingly complex network environment, presenting new challenges to network security. Maintaining network security and stability is becoming increasingly difficult, as malicious software, terminal platforms, and network protocols become more complex, making it easier to expose network vulnerabilities [1].

Traditional network security techniques mainly rely on signature technology to detect network attacks, but they are increasingly ineffective against “zero-day attacks”. A zero-day attack refers to a network attack that exploits vulnerabilities that software developers are not aware of before they are discovered by attackers. Due to the lack of effective protection measures, these vulnerabilities are easily exploited by attackers, resulting in system and data damage or leakage, as well as consequences such as data breaches, system failures, and financial losses. Signature-based network security technology often adds attack signatures to the database only after serious consequences of an attack have occurred [2].

In 2021, a remote code execution vulnerability in Apache Log4j2 based on the Java language was publicly disclosed, affecting multiple systems and applications worldwide that rely on Apache Log4j2 [3]. If successfully exploited, attackers can remotely execute arbitrary code on vulnerable systems, potentially leading to serious consequences, such as data leakage, system crash, unauthorized access to sensitive information, and more.

In order to cope with these challenges, traffic anomaly detection, as an effective network security protection and anomaly detection technology, has been widely applied and researched [4,5,6,7].

The first type is classification algorithms based on statistical methods. These algorithms attempt to model behavioral indicators by using single or multiple variables [8]. The second type is the classification algorithm based on domain knowledge [9]. The third category is classification algorithms based on machine learning methods [10]. At present, the main research focus of anomaly detection technology is on traditional machine learning and deep-learning-based anomaly detection [11].

In the field of traditional machine learning, Bhattacharya et al. [12] proposed a principal component analysis firefly-based XGBboost classification model, which achieved good performance using an open-source dataset from Kaggle. Ding et al. [13] proposed a mixed intrusion detection method based on K-nearest-neighbors (KNN) and generative adversarial networks for imbalanced data. Balyan et al. [14] inherited the enhanced genetic algorithm, particle swarm optimization (EGA-PSO) and improved random forest (IRF) method, established an efficient detection model based on hybrid network, and achieved high accuracy in the NSL-KDD dataset. Ahmim et al. [15] showed certain advantages in terms of false positive rate and time cost by combining decision trees with different classification algorithms. Tao et al. [16] proposed FWP by combining the characteristics of genetic algorithm and vector machine algorithm, and designed a fitness function. Aljanabi et al. [17] proposed an improved binary classification algorithm based on the TLBO teaching and learning algorithm and genetic algorithm, which showed good performance on the KDDCUP dataset. Ioannou et al. [18] designed a detection algorithm based on binary metalogic regression (BLR), taking local node parameters of benign and malicious behaviors as input, and achieved attack detection accuracy in the range of 96–100%. Cao et al. [19] combined random forest algorithm and Pearson correlation analysis to select features and solve the problem of feature redundancy. Then, using convolutional neural networks to extract spatial features has better accuracy in similar models. Ding et al. [20] proposed a real-time anomaly detection algorithm LGMAD based on short-term memory and the Gaussian mixture model, and achieved good performance on self-made datasets.

Suda et al. [21] proposed an anomaly detection method based on RNN for the usage scenario of vehicular networks. This method can extract time-series features of network packets and can detect various types of traffic anomalies, such as ID spoofing and flooding attacks. Roy et al. [22] proposed an anomaly detection method using bidirectional long short-term memory recurrent neural networks (BLSTMRNNs) for network traffic detection scenarios in the IoT. They studied the performance of binary classification between normal and attack traffic in the IoT. Li et al. [23] proposed an unsupervised multivariate anomaly detection method based on generative adversarial networks (GANs), and created two new datasets to effectively validate the model. Min et al. [24] proposed a network intrusion detection method using a memory-augmented deep autoencoder (MemAE). The MemAE model is trained to reconstruct the input of an abnormal sample that is close to a normal sample, which solves the generalization problem for such abnormal samples. It was confirmed that the proposed method is better than other one-class models through experiments conducted on the NSL-KDD, UNSW-NB15, and CICIDS 2017 datasets. Xu et al. [25] proposed TabTransformer based on the self-attention mechanism and gradient lifting decision tree, which can effectively extract discrimination features and achieve good results in NAD data sets.

At present, most of the research on network traffic anomaly detection is based on existing open source data sets, and network traffic detection is carried out in different scenarios and conditions. In conclusion, the model based on the short- and long-term memory network can effectively preserve the long-term dependence relationship between traffic data. The model based on autoencoder can be used as a kind of data-processing method, and can also be used as a classifier to detect data. The model based on GANs is more suitable for small sample model training. A model based on transformer can use the self-attention mechanism to more fully extract the context of the data and accurately capture the abnormal relationship between the data.

Firstly, this article investigates the construction process of a traffic anomaly detection dataset, which involves constructing features from network packets and transforming them into a traffic anomaly detection dataset, thereby providing data support for algorithm training. The current classic dataset is either older or lacks comprehensive feature extraction; the existing open-source feature extraction tools have some errors and cannot meet the demand for extracting more features. Based on the above two points, the feature extraction program was re-developed on the Linux platform, and a more comprehensive set of features was designed by combining domain knowledge and different machine learning algorithms so that abnormal traffic types can be distinguished through this set of features, and then feature elimination can be performed to reduce the redundancy between features. Through experiments, it was shown that the entire feature-processing process proposed in this article can reduce the time and spatial complexity of the algorithm and improve its performance. Secondly, a network model based on improved RNN and the self-attention mechanism was proposed. Network traffic data are time series data, and RNN, as one of the best benchmark models for solving sequence problems in deep learning, still has some problems, such as the long-term dependence of RNN, where prediction results rely on information from a long time ago. The short- and long-term memory network and gated recurrent neural network are improved models to solve the problems of gradient disappearance and gradient explosion of RNN. This article uses an N-layer improved RNN model to learn temporal information in network traffic data, but the importance of different information in the sequence is different. The self-attention mechanism can effectively learn the “noteworthy” parts of the sequence. In order to verify the effectiveness of the proposed network model, it achieved good results compared to the benchmark model on the constructed dataset.

The entire article is divided into three main parts: the first part is the creation of a feature-based engineering dataset, the second part is the anomaly network traffic detection method based on self-attention networks, and the third part is a summary of the previous content and future prospects.

## 2. Creating Datasets Based on Feature Engineering

In 1998, the US Department of Defense Advanced Planning Agency simulated the network environment of the US Air Force in an intrusion detection project, using various types of traffic for training and testing. After preprocessing, this dataset formed the KDDCUP99 dataset [26], providing a unified performance evaluation benchmark for future network traffic anomaly detection systems, However, the KDDCUP99 dataset is still quite rough, containing a large amount of redundant and duplicate data, with a significant deviation in the proportion of normal and abnormal data. In response to these issues, Tavallaee et al. improved KDDCUP99 to form the NSL-KDD dataset [27]. After entering the 21st century, the popularity of network technology has grown exponentially, and the rapid advancement of infrastructure and software facilities has led to drastic changes in the network environment. Due to the age of datasets, such as NSL-KDD, there are significant differences in network traffic distribution compared to recent years. The types of attacks are constantly evolving, and with the development of hardware technology, collecting information and obtaining datasets have become easier than before. Datasets with richer traffic characteristics and more reasonable data distribution are also constantly emerging, such as CICDS2017 [28]. In addition, there are datasets targeting certain types of attacks, such as DDoS 2019 [29], and datasets targeting distributed denial-of-service attacks. In addition, different fields will also develop their own traffic anomaly detection datasets based on the characteristics of the business they are responsible for, although these results will not be easily disclosed.

UNSW-NB15 is a dataset on network security used to evaluate the performance of network attack detection systems [30]. It is provided by the School of Computer Science at the Australian National University (UNSW). This dataset contains a large amount of simulated network traffic data, including various types of network attacks, such as denial of service (DoS) attacks, Trojan attacks, and malware attacks. The dataset includes 49 features, most of which are technical indicators of network traffic protocols, such as packet size and time intervals. This dataset is very popular in the field of network security research and is widely used to test the performance of network security algorithms.

However, these features cannot accurately describe the characteristics of network traffic. For example, a key indicator for identifying DoS attacks is the frequency of sending data packets in a short period of time, the key indicator for identifying SYN flooding attacks is the number of SYN in the TCP header, and the auxiliary indicator for identifying XSS injection is the difference in size between different packets in a session. These key features are not included in the UNSW-NB15 dataset, which limits the detection capability of algorithms.

This article proposes to add the computation of many statistical indicators to the UNSW-NB15 feature set, redesign the feature set, and construct a dataset that can help improve the algorithm’s detection performance. The overall process is shown in the Figure 1.

### 2.1. Constructing Datasets

#### 2.1.1. Perform Feature Extraction

The first step is to read the pcap file in binary mode and extract the link layer data frames from it. After reading the link layer data frame into the computer memory, proceed to the next step of processing. In addition, the program implemented in this article does not currently support the IPv6 protocol, so it is necessary to filter out the data frames of the IPv6 protocol in this step.

The second step is to segment and reassemble the extracted link layer data frames. Link layer sharding refers to dividing large data packets into several small data packets, which are processed separately during transmission. It usually occurs at the link layer, which is the second layer of network data transmission. The reason for sharding is that in the network, the size of the data packets may exceed the maximum value transmitted by the network, so sharding is necessary for transmission. The fragmented data packets can be recombined in order at the destination to form the original big data packets. Link-layer sharding not only improves the transmission efficiency but also helps prevent network data loss. When network congestion occurs, fragmented data packets can be more easily transmitted through the network, thereby reducing the risk of data loss. Therefore, several shards should belong to the same network layer data packet. In this step, based on the header of the link layer data frame, different shards belonging to the same data packet are identified and reorganized into network layer data packets; for data frames that have not been sharded, the header of their frames is directly removed. The result of this step is several network layer packets.

The third step is to aggregate network layer packets belonging to the same session. Here, a session refers to a series of peer-to-peer communication operations between two network devices. They establish connections by sending information to each other to complete specific tasks, such as file transfers or database queries. This article does not separate feature extraction for each network layer packet, but rather aggregates it as a session. The reason for doing this is that a continuous session may contain an extremely large number of network layer packets, and the information of each packet is highly correlated. Extracting the features of each network layer packet separately often results in overly discrete information. In addition, it is also very difficult for the algorithm to recognize different data packets that belong to the same session but are scattered in order, and the time-series information in the dataset is also difficult to embed into the algorithm model.

For TCP protocol packets, since TCP itself is a connection-oriented transport layer protocol, each connection ends with three handshakes and four waves; the packets in each connection can be considered the same session. For UDP and ICMP protocol packets, there is no concept of connection, and this article uses a timeout mechanism for determination. When A and B send data packets for the first time, set a timer and timeout for them. If the data packets are sent again between A and B before the timeout, reset the timer. If A and B do not send any more packets until the timeout, it is considered that the packet from the first packet to the one before the timeout belongs to the same session. According to the protocol fields in the IP protocol header, there are theoretically 256 protocols. However, the other 253 protocol types have a very low frequency in most network environments, so this article will not consider these protocol types of packets for the time being and will directly filter them out.

#### 2.1.2. Feature Design and Selection

Feature extraction refers to extracting meaningful features from raw data for further analysis and processing. Feature extraction is often a preprocessing step in machine learning algorithms, aiming to select useful information from raw data to make it easier for the algorithm to identify patterns in the input data. In order to comprehensively and accurately describe network traffic features, this article designs five categories of features based on the extracted binary session data, as shown in the Table 1.

Here, the sender of the first data packet in each session is defined as the client, and the receiver is defined as the server.

The first category of features is the identity features of the session. There are a total of five identity features, which are the network addresses and communication protocol types of the two network devices in each session.

The second category of features is transmission features. There are a total of 38 transmission features, which aim to describe the statistical characteristics of data packet transmission within each session.

The third category of features is TCP features. There are a total of 27 TCP features, which aim to describe the statistical characteristics of the fields contained in the TCP sessions. If it is a non-TCP session, it is simply set to 0.

The fourth category of features is frequency features. There are a total of 19 frequency features, which aim to describe the frequency information of data packet transmission within each session.

The fifth category of features is duplicate features. There are a total of 19 duplicate features, which aim to describe the statistical characteristics of multiple sessions within a period of time.

There are a total of 108 features, which can accurately describe the state information of the sessions from various perspectives. Some features are shown in the Table 2. Obviously, some of these features are redundant or correlated. However, on the one hand, this article believes that providing direct information to the detection algorithm is a better choice than letting the algorithm dig out hidden information from the data itself. On the other hand, this article will also perform feature selection to identify the best subset of features.

In addition, each session has a label feature, which is the category of the session. If there is no manual labeling or given labeling rules, this feature is empty.

The dataset constructed in this article has multiple types of features, and some of the features do not follow normal distribution. Therefore, a combination of filter and wrapper methods is used for feature selection.

Firstly, features that cannot be used for model training, namely the four categorical features (client_ip, client_port, server_ip, server_port) and two timestamp features (start_time, end_time) are removed. The meanings of these features are already reflected in the session aggregation and the sequence of sessions.

For categorical features, such as protocol, service, and state, since their values do not have a sense of size and order and the number of feature values is not large, one-hot encoding is performed. This means adding a binary feature for each discrete value variable.

When there are a large number of categories, the one-hot encoding vector can become very large, causing memory and computational difficulties. Therefore, in this experiment, a recursive feature elimination process is designed to filter out redundant, unrelated, and low-information features to improve model efficiency and performance.

For the categorical label feature type, it is converted into 0 and 1 according to normal and abnormal, respectively.

All other numerical features are standardized. Then, according to the filter method, the variance of each feature is calculated, and features with a variance smaller than $10^{-5}$ are removed. This yields 121 features for selection.

Finally, using the wrapper method, all numerical features, except type, are used as the initial feature set, with type as the label feature. Machine learning algorithms, such as stochastic gradient descent classification and XGBoost, are used, with the F1-score as the evaluation metric. Recursive feature elimination with cross validation is performed on the feature set, and the trend of the F1-score is observed under different feature set sizes to select the best feature set.

Here, recursive feature elimination is a method of evaluating the importance of features to the model by recursively deleting features. The specific steps are shown in Algorithm 1.
**Algorithm 1:** Recursive feature elimination algorithm**Input**: classification data set *D* with *n* features; basic classification algorithm model (such as random forest, logistic regression, etc.); the number of features *k* (1≤k≤n) for iteration stop; the number of features eliminated at each step.1Initial feature set $F=\{f_{1},f_{2},\ldots ,f_{n}\}$2Training the basic algorithm model:use the *F* feature set of the data set *D* to train the basic algorithm model;3Calculate feature importance: for each feature in *F*, calculate its impact on model prediction results according to the training results of the model.4Recursively delete features: delete the least important step features in *F*;5Repeat steps 2 to 4 until the required number of features is reached, i.e., |F|≤k;6Preserve the selected features: use the feature set F to train the final model, and output the feature subset F and model.**Output**: A feature subset composed of *k* features.

In order to reduce the impact of randomness, K-fold cross validation is used, which means that K-1 parts are used for training and 1 part is used for testing each time. After K training sessions, the test results are averaged.

### 2.2. Experimental Results and Analysis


**Experimental Results:**


The original data for feature extraction in this article come from the UNSW-NB15 dataset’s pcap packets. After processing with the feature-extraction program in this article, a dataset of 2,445,079 records is obtained.The experimental data results are shown in the Table 3. The specific network traffic types in the dataset are shown in the Table 4.

First, classic machine learning models are trained on the dataset as a benchmark performance. The dataset is divided into a training set, validation set, and test set in an 8:1:1 ratio. Then, linear classifiers, decision trees, random forests, LightGBM, XGBoost, and multilayer perceptron are trained and tested, and the results are shown in the Table 5.

As the random forest has the best overall performance, it is used as the main reference model for feature selection. Therefore, the dataset is divided into training and test sets in an 8:2 ratio, and the recursive feature elimination with 5-fold cross-validation is performed on the training set using the random forest algorithm, with the F1-score as the evaluation metric, and the effect is evaluated on the test set. The final test set effect under different feature set sizes is shown in the Figure 2.

When the number of features in the subset was 19, the algorithm achieved the best F1-score of 0.6272. Therefore, 19 features were selected as the final optimal subset from the original 121 features.

The dataset was then split into training, validation, and testing sets with a 8:1:1 ratio.The experimental results are shown in the Table 6. To evaluate the impact of the feature subset on algorithm performance, the model was retrained using only the 19 selected features, and the testing results were compared to those using all features as shown in Table 7. It can be observed that the random forest algorithm still achieved the best F1-score of 0.6786, which is 2.1% higher than before feature selection. This indicates that feature selection effectively improves the overall performance of the algorithm.

In order to show the relationship between each feature dimension, this article selects some eigenvalues and obtains the feature correlation matrix by calculating the correlation coefficients between different features. The thermal map is shown in Figure 3. The closer the correlation coefficient of two features is to 1, the greater their positive similarity; conversely, the closer the correlation coefficient is to −1, the greater the reverse similarity. Additionally, the darker the color, the stronger the correlation between the two features. The characteristics of service_diff_host_ratio and service_host_same_src_prot_ratio shown in the figure are very similar. As can be seen from the figure, the features selected in this article are relatively independent.

Based on the above, this chapter reconstructs a dataset for network traffic anomaly detection (DNTAD). We use the model presented in the next section.

## 3. Self-Attention-Based Detection Algorithm

In traffic anomaly detection tasks, each session is described by a one-dimensional vector containing several features in the form $s=s_{1},s_{2},s_{3},\ldots ,s_{m}$, where m is the number of numerical features, $s_{i}\in R$.

Each session corresponds to an exception type y∈N. For example, in a binary classification scenario, *y* = 0 represents normal network traffic, and *y* = 1 represents abnormal traffic; in a multi-classification scenario, *y*≠ 0 represents the *y*-th abnormal type, that is, a specific abnormal type.

Therefore, the constructed training data set is an ordered set composed of a series of session feature vectors $S=\{<s_{1},t_{2}>,<s_{2},t_{2}>,\ldots ,<s_{n},t_{n}>\}$, where $t_{1}\leq t_{2}\leq t_{3}\leq \cdots \leq t_{n}$, and $t_{i}$ represents the timestamp of the *i*-th session feature vector, that is, the session feature vectors are arranged in chronological order. *n* is the total number of vectors. Since each feature vector corresponds to a label feature, i.e., anomaly type *y*, the labeled training dataset can be expressed as D=〈S,y〉.

The dimension of *S* is n×(m+1), $y=\{y_{1},y_{2},y_{3},\ldots ,y_{n}\}$. Each $s_{i}$ corresponds to $y_{i}$.

Given a labeled anomaly detection dataset *D*, the task of an anomaly detection algorithm is to learn the functional correspondence F between S and y through methods such as neural networks F:y = f(s).

### 3.1. Network Architecture Design

In this article, a deep neural network model LS is designed based on the structure of the cyclic neural network, self-attention mechanism, convolutional neural network and fully connected layer. The complete model structure based on self-attention is shown in Figure 4.

In a network traffic sequence data, each feature has different meanings and importance, so it is necessary to fully explore the importance of each feature. In the learning model that adopts the attention mechanism, the attention mechanism is usually placed in the latter part of the whole structure, and the hierarchical features of the input sequence are extracted through the network learning of the former part, and then the key abstract information is extracted by the attention mechanism.

Session: Session data with a series of features that have been normalized before input. batch_size sessions are inputted at the same time, each with n_feature features.

RNN layer: A recurrent neural network layer. This layer usually uses LSTM or GRU to handle sequential data to capture long-term dependencies. The input matrix has a dimension of (batch_size, n_feature), and the output matrix has a dimension of (batch_size, hidden_size), where hidden_size is the dimension of the hidden state. The more dimensions there are, the more information the algorithm can learn, but the learning difficulty is also higher.

Masked multi-head attention: A masked multi-head self-attention layer. This layer uses a multi-head self-attention mechanism to identify important information at each time step and generate a weight vector that represents the importance of each time step. The input is (batch_size, hidden_size) and the output is (batch_size, hidden_size):(1)$$MaskedMultiHead\left(X\right)=Concat\left(head\_1,head\_2,\ldots ,head\_h\right)W^{O}$$
(2)$$head\_i=MaskedSelfAttention(XW_{i}^{Q},XW_{i}^{K},XW_{i}^{V},Mask)$$

Layer normalization: A commonly used method for layer normalization. It first calculates the mean and standard deviation of the output data of each layer, and then standardizes each data item in the layer to obtain the standardized data as the output of the layer.

Convolutional layer: A one-dimensional convolutional neural network is a special type of convolutional neural network that is suitable for processing one-dimensional signals and performs convolution only along one dimension. The pooling layer mainly reduces the dimensionality and downsamples the input data to reduce the size of the network and prevent overfitting. Here, the maximum pooling method is used, which selects the maximum value in a fixed-sized pooling window as the output.

Fully connected layer: This layer is the output layer of the network, which uses a weight vector to weight the output of the pooling layer and obtain the final context vector for prediction. The input dimension is (batch_size, seq_len, hidden_size), and the output dimension is (batch_size, output_dim), where output_dim is the dimension of the prediction results.

The add operation here refers to the residual connections. Residual connections refer to adding the input data to the output data of the current layer in the hidden layers of the neural network. Specifically, in each hidden layer, the input data are first subjected to a series of linear and nonlinear transformations, and then the output of the layer is added to the input data to obtain the final output of the layer. This connection method allows the model to pass gradients more quickly by skipping some layers and can help the network better capture high-dimensional features in the data.

### 3.2. Experimental Results and Analysis

#### 3.2.1. Experimental Setup

Use the dataset presented in the previous section. The dataset is divided into a training set, verification set and test set according to the ratio of 8:1:1. The training set is used for algorithm iteration, and the effect is checked on the verification set after each round of iteration. When the effect of the verification set for multiple consecutive iterations is no longer improved or decreased, the iteration is stopped, and the model with the best performance on the verification set is used as the final model. We test the model, and make predictions on the test set as the final model performance.

The model uses cross entropy as the loss function. Its formula is as follows:(3)$$L=-\sum_{i=1}^{N}y^{\left(i\right)}\log\hat{y^{\left(l\right)}}+\left(1-y^{\left(i\right)}\right)\log\left(1-\hat{y^{\left(l\right)}}\right)$$
where *N* is the number of samples in the dataset, *y* is the one-hot encoding of the true label, and $\hat{y}$ is the probability distribution of the predicted outcome. For the prediction result of each sample in the classification task, the cross-entropy loss function is used to measure the gap between the prediction result and the real result. In addition, due to the large gap in the proportion of category distribution in this experiment, in order to make the model more sensitive to abnormal samples, this experiment assigns different weights to different categories. In the case of binary classification, it can be expressed as
(4)$$L=-\sum_{i=1}^{N}w_{0}y^{\left(i\right)}\log\hat{y^{\left(l\right)}}+w_{1}\left(1-y^{\left(i\right)}\right)\log\left(1-\hat{y^{\left(l\right)}}\right)$$

Here, $w_{0}$ is the sample weight for y=0, and $w_{1}$ is the sample weight for y=1. The greater the sample weight, the greater the penalty to the algorithm when the algorithm predicts wrongly. Therefore, setting a larger value for $w_{1}$ can make the algorithm more sensitive to samples of y=1. Considering that there are only one-fifth of the samples of y=1 in the data set, such a setting helps to improve the overall effect of the algorithm.

The Adam optimization algorithm is used when optimizing the model, which adapts the learning rate of each parameter by maintaining the exponential moving average of the squared gradient of each parameter (first moment) and the exponential moving average of the original gradient (second moment). Its calculation rules at step *t* iteration are as follows:(5)$$\begin{array}{cc} m_{t} & =\beta_{1}*m_{t-1}+\left(1-\beta_{1}\right)g_{t} \end{array}$$(6)$$\begin{array}{cc} v_{t} & =\beta_{2}*v_{t-1}+\left(1-\beta_{2}\right)g_{t}^{2} \end{array}$$(7)$$\begin{array}{cc} \theta_{t} & =\theta_{t-1}-\alpha m_{t}/\left(\sqrt{vt}+\epsilon \right) \end{array}$$

Here, $\beta_{1}$ and $\beta_{2}$ are hyperparameters, $0<\beta_{1}<1,0<\beta_{2}<1$; $m_{t}$ is the exponential moving average of the gradient of the parameter; $v_{t}$ is the exponential moving average of the gradient square; α is the learning rate; ϵ is a very small positive number, used to prevent the denominator from being 0; $g_{t}$ is the current gradient; and $\theta_{t}$ is the value of the current parameter. This article sets the initial learning rate to 0.005, and the learning rate increases to 60% every 5 epochs.

#### 3.2.2. Experimental Results

In order to horizontally compare the performance of the LS model proposed in this article with other algorithm models, this chapter selected the commonly used XGBoost iterative algorithm, random forest algorithm in the field of traffic anomaly detection, and the recently proposed hybrid network by Atifi et al. [31] as the control group for comparative experiments. According to the Table 8, it can be seen that the LS model proposed in this article has the best F1-score on the DNTAD dataset, reaching 74.56%. Compared to models such as decision tree, random forest, LightGBM, XGBoost, LSTM, GRU, Hybrid, etc., it is improved by 14.04%, 6.7%, 11.54%, 11.05%, 3.05%, 4.14%, and 1.86%, respectively.

In order to more intuitively reflect the performance comparison results, this chapter visualizes the comparison results and draws the F1-score of each model on the DNTAD dataset, as shown in the Figure 5.

To evaluate the contribution of each model component, this article designs a series of ablation experiments. Under the settings of different normalization methods, different numbers of LSTM layers, and different numbers of self-attention heads, the test results after training are shown in the table. These data show that the standardization method, residual structure, multi-head self-attention and other components in Table 9 have effectively improved the performance of the model.

Two standardization methods are used in this article. First, the optimal F1-score is 0.7437 without using the standardized method. In the case of layer standardization, the optimal F1-score is 0.7456; with batch standardization, the optimal F1-score is 0.7219. Therefore, the best choice is to use a layer standardization approach.

In this article, two kinds of cyclic neural network models—LSTM and GRU—are tested. Based on the LSTM model, the optimal F1-score of single layer LSTM is 0.7392. The best F1-score for the two-tier LSTM was 0.7456. Therefore, the best choice is to use a two-layer LSTM.

The best F1-score of one self-attentional head was 0.7404; the best F1-score of two self-attentional heads was 0.7456; the best F1-score of four self-attentional heads was 0.7424; and the best F1-score of eight self-attentional heads was 0.7409. Therefore, the best choice is to use two self-attention heads. Select the optimal model settings, and the test results under different types of weight settings are shown in Table 10.

When the class_weight is $w_{0}=1.0$ and $w_{1}=2.0$, the maximum F1-score is 0.7456. Therefore, set class_weight to $w_{0}=1.0$ and $w_{1}=2.0$. Select the optimal model settings, and the test results under different types of weight settings are shown in the table.

Selecting the optimal model setting and weight set and replacing LSTM with GRU, the F1-score obtained in the test is 0.7377, which is lower than the optimal model performance, hence the choice to use LSTM.

The above experiments show that the self-attention deep learning algorithm designed in this article can have better detection results than classic machine learning and deep learning algorithms.

This article introduces the deep learning algorithm model designed according to the structure of the convolutional neural network, recurrent neural network, and self-attention mechanism. After training and testing on the data set, it is verified that the deep learning algorithm is significantly better than the classic machine learning algorithm. This chapter also designs ablation experiments to verify the effectiveness of each component in the algorithm model.

## 4. Conclusions

This article uses the raw data of the classic data UNSW-NB15, which provides a rich range of network scenarios and attack types. A feature extraction program was written using libpcap based on the UNSW-NB15 dataset. On the one hand, it can analyze the structure of pcap files and extract link layer data frames from them; on the other hand, it can also directly capture real link layer data frames from network device drivers. Using this program, this article aggregates traffic belonging to the same session, calculates a series of characteristics of the session, and thus reconstructs a network traffic anomaly detection dataset.

Secondly, this article designs a relatively complete feature set for network traffic at different levels and protocol types, with a total of 108 items in 5 categories, which can accurately describe the state information of the session. Due to the large number of these features and the presence of invalid or redundant features, feature selection was performed on the feature set. This article selects a 19-dimensional feature subset through filtered feature selection and feature recursive elimination methods. Through the validation of machine learning algorithms, it is shown that the dataset construction method for network traffic in this paper can effectively reduce the algorithm training time and improve the algorithm detection ability when using the same machine learning algorithm and raw traffic data.

Subsequently, this article designs a deep learning network detection model based on the autonomous attention mechanism, which effectively improves the efficiency and accuracy of network traffic processing. After training and testing on the dataset constructed in this article, it was found that its detection performance is significantly superior to classic algorithms and recent models in the field of network traffic detection. Moreover, through a series of ablation experiments, this article demonstrates that each module in the model has a significant improvement in detection accuracy.

## 5. Prospect

The current model only supports binary scenarios. The algorithm designed in this article is targeted at the detection scenario, that is, it only needs to identify the session as “normal” or “abnormal”, but it can be considered to transfer it to the identification scenario, that is, to identify specific types of exceptions. The UNSW-NB15 dataset actually provides nine exception-type annotations of Fuzzers, Analysis, Backdoors, DoS, Exploits, Generic, Reconnaissance, Shellcode and Worms. However, the data amount of the different exception types varies greatly, resulting in a poor classification effect. Therefore, it is necessary to improve both the data set and the classification algorithm to obtain a better recognition algorithm.

## Figures and Tables

**Figure 1 sensors-23-05059-f001:**
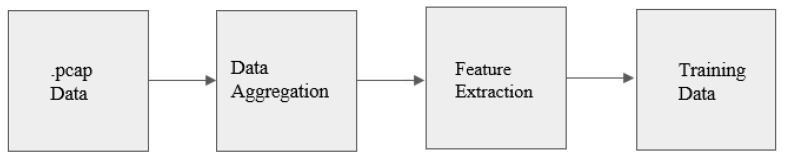
Data-processing workflow.

**Figure 2 sensors-23-05059-f002:**
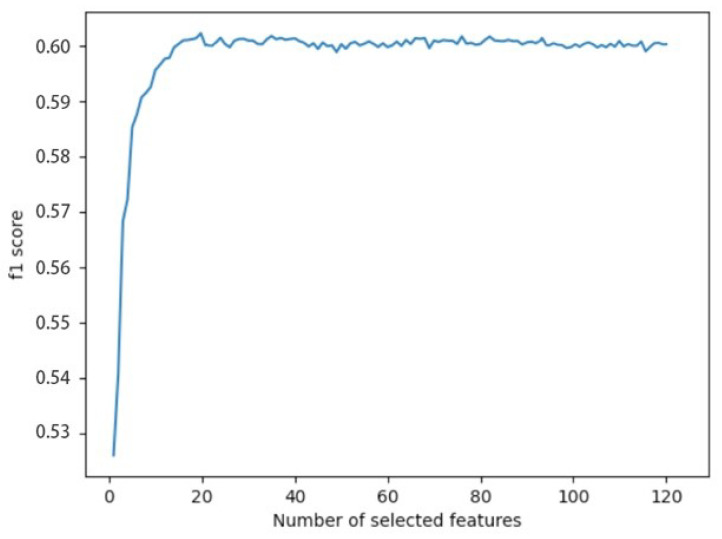
F1-score of feature selection process.

**Figure 3 sensors-23-05059-f003:**
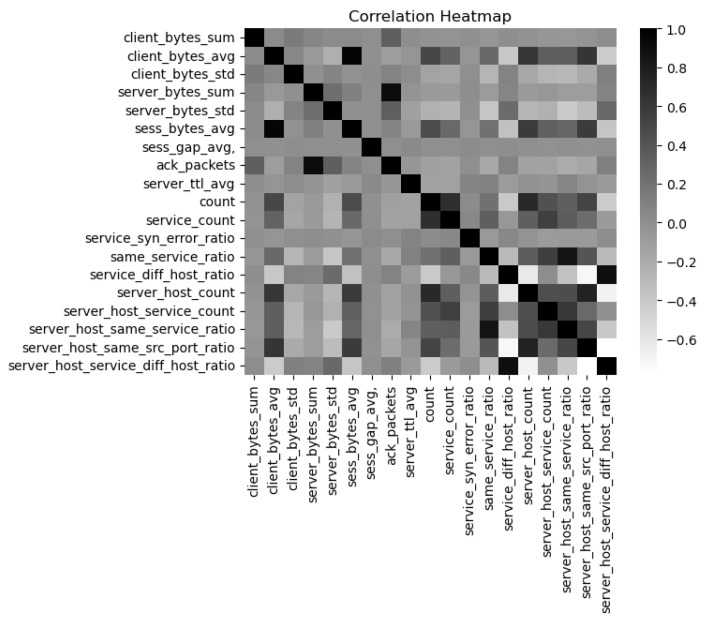
Heat maps of partial features.

**Figure 4 sensors-23-05059-f004:**
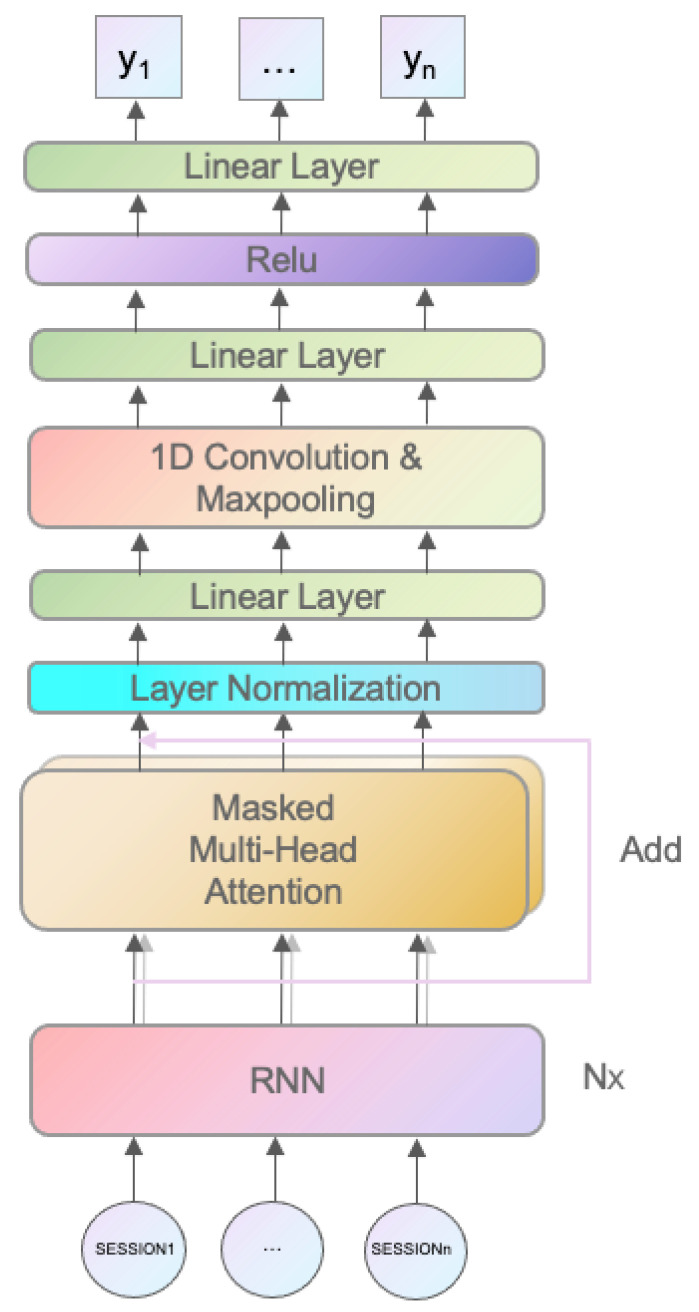
Neural network architecture.

**Figure 5 sensors-23-05059-f005:**
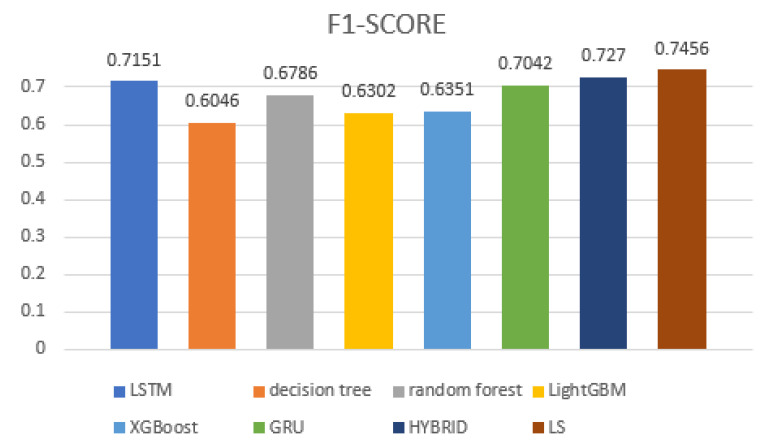
Comparison of F1-scores between different models.

**Table 1 sensors-23-05059-t001:** All of the feature types.

Feature Category	Quantity
Identity features of conversation	5
Transmission features of the session	38
Frequency features of session	19
The repetitive features of session	19
TCP session features	27

**Table 2 sensors-23-05059-t002:** Some of the features.

Feature Category	Quantity
client_ip	IP address of the client
server_ip	IP address of the server
protocol	IP protocol type
client_packets	Total number of packets sent by the client
client_bytes_avg	The average number of bytes sent by the client in packets
src_init_window_bytes	Number of bytes sent at initial window size
client_ack	The number of ACK flags in TCP sent by the client
syn_error_ratio	The percentage of SYN errors in sessions where the server IP is consistent with the current session in the past 2 s
rej_error_ratio	The percentage of REJ errors in sessions where the server IP is consistent with the current session in the past 2 s

**Table 3 sensors-23-05059-t003:** Number and percentage of different session types in the new dataset.

Session Type	Quantity	Percentage
Normal	1,916,035	75.51%
Abnormal	524,544	21.49%

**Table 4 sensors-23-05059-t004:** Number of different session types in the DNTAD dataset.

Session Type	Quantity
Normal	1,916,035
Generic	340,502
Exploits	76,088
Fuzzers	42,143
Dos	29,223
Reconnaissance	24,697
Analysis	4753
Backdoors	4140
Shellcode	1687
Worms	311

**Table 5 sensors-23-05059-t005:** Machine learning algorithm training and test results.

Algorithm		Linear Classfication	Decision Tree	Random Forest	LightGBM	XGBoost	Multilayer Perceptron
Training set	accuracy	0.7772	0.9931	0.9930	0.8664	0.8818	0.8570
	precision	0.3123	0.9680	0.9809	0.5602	0.5167	0.4641
	recall	0.4445	0.9987	0.9851	0.7305	0.8538	0.7483
	F1-score	0.3669	0.9831	0.9830	0.6342	0.6438	0.5729
Testing set	accuracy	0.7786	0.8372	0.8776	0.8651	0.8782	0.8561
	precision	0.3118	0.6019	0.5586	0.5534	0.4999	0.4542
	recall	0.4428	0.6028	0.7820	0.7233	0.8410	0.7438
	F1-score	0.3659	0.6024	0.6517	0.6270	0.6271	0.5640

**Table 6 sensors-23-05059-t006:** Machine learning algorithm training and test results after feature selection.

Algorithm		Linear Classification	Decision Tree	Random Forest	LightGBM	XGBoost	Multilayer Perceptron
Training set	accuracy	0.7027	0.9928	0.9927	0.8663	0.8830	0.8513
	precision	0.4940	0.9666	0.9806	0.5546	0.5229	0.40291
	recall	0.3452	0.9983	0.9837	0.7313	0.8518	0.7634
	F1-score	0.4064	0.9822	0.9822	0.6308	0.6480	0.5275
Testing set	accuracy	0.7025	0.8354	0.9047	0.8660	0.8787	0.8483
	precision	0.4953	0.6043	0.5845	0.5482	0.5067	0.3893
	recall	0.3490	0.6049	0.8089	0.7409	0.8506	0.7676
	F1-score	0.4095	0.6046	0.6786	0.6302	0.6351	0.5166

**Table 7 sensors-23-05059-t007:** Machine learning algorithm training and test results before and after feature selection.

Algorithm	Linear Classification	Decision Tree	Random Forest	LightGBM	XGBoost	Multilayer Perceptron
Before feature selection	0.3659	0.6024	0.6517	0.6270	0.6271	0.5640
After feature selection	0.4095	0.6046	0.6789	0.6302	0.6351	0.5166

**Table 8 sensors-23-05059-t008:** LS model compares other algorithms in DNTAD dataset.

Model	F1-Score
LSTM	0.7151
decision tree	0.6046
random forest	0.6786
LightGBM	0.6302
XGBoost	0.6351
GRU	0.7042
Hybrid	0.727
LS	0.7456

**Table 9 sensors-23-05059-t009:** Algorithm performance under different experimental settings.

Standardized Method	LSTM Layers	Number of Heads	Accuracy	Precision	Recall	F1-Score
None	1	1	0.9792	0.6789	0.8093	0.7384
		2	0.9790	0.7249	0.6812	0.7024
		4	0.9793	0.6805	0.8089	0.7392
		8	0.9797	0.7016	0.7675	0.7331
	2	1	0.9797	0.6862	0.8117	0.7437
		2	0.9797	0.6886	0.8058	0.7426
		4	0.9798	0.6899	0.8036	0.7424
		8	0.9796	0.6870	0.8040	0.7409
Add and LayerNorm	1	1	0.9794	0.6859	0.7944	0.7362
		2	0.9793	0.6845	0.7942	0.7353
		4	0.9795	0.6897	0.7894	0.7362
		8	0.9799	0.7091	0.7596	0.7335
	2	1	0.9798	0.6928	0.7950	0.7404
		2	0.9800	0.6926	0.8074	0.7456
		4	0.9799	0.7191	0.7357	0.7273
		8	0.9797	0.6976	0.7767	0.7350
Add and BatchNorm	1	1	0.9781	0.6596	0.8078	0.7262
		2	0.9761	0.6220	0.8404	0.7149
		4	0.9738	0.5863	0.8712	0.7009
		8	0.9779	0.6583	0.7991	0.7219
	2	1	0.9788	0.7115	0.6987	0.7050
		2	0.9793	0.7205	0.7034	0.7118
		4	0.9799	0.7395	0.6955	0.7168
		8	0.9796	0.7186	0.7216	0.7201

**Table 10 sensors-23-05059-t010:** Performance of the algorithm under different experimental settings.

Class Weight	Accuracy	Precision	Recall	F1-Score
$w_{0}=1.0,w_{1}=1.0$	0.9821	0.7657	0.6608	0.7094
$w_{0}=1.0,w_{1}=2.0$	0.9820	0.6926	0.8074	0.7456
$w_{0}=1.0,w_{1}=3.0$	0.9807	0.6693	0.8202	0.7371
$w_{0}=1.0,w_{1}=4.0$	0.9806	0.6669	0.8198	0.7355

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
