# Peer review of "Research on Anomaly Network Detection Based on Self-Attention Mechanism"

_sensors, 2023, doi:10.3390/s23115059_

Round 1

Reviewer 1 Report

The paper is adequately structured and discusses a worthwhile topic. While the use of AI techniques for network traffic analysis is worthwhile, the paper's contributions should be considerably improved. First of all, the key sections of results discussion and threats to validity are missing. Furthermore, a detailed comparison of obtained results should be carried out against existing, up to date literature -- this should also improve the references section, which is currently underwhelming. Another option is to publish the obtained results as an open data package that other researchers could use to replicate or extend these experiments. In addition, I've also noted a few minor observations regarding paper structure/contents:

  • The last phrase of the abstract mentions that the benchmark model was improved "by 6.7", but measurement units are not provided
  • Regarding the features presented in Section 2.1.2, they should be elaborated upon or at least, exemplified
  • Paragraph "Firstly, features that cannot be used for model training" is repeated. Likewise for paragraph starting "For categorical features protocol, …"
  • Both Section 4 and 5 are titled Conclusion
  • Some words are repeated in Algorithm 1

Use of English is fine.

Author Response

Dear Reviewers,

Thank you very much for your time involved in reviewing the manuscript and your very encouraging comments on the merits.

We appreciate your clear and detailed feedback and hope that the explanation has fully addressed all of your concerns. In the remainder of this letter, we discuss each of your comments individually along with our corresponding responses.

To facilitate this discussion, we first retype your comments in italic font and then present our responses to the comments

Reviewer 2 Report

Overall the quality of English is satisfactory.

However, the whole article must be reviewed carefully in terms of typos, grammatical errors and sentence structure at various places. 

Author Response

(The authors gave the same response as above.)

Round 2

Reviewer 2 Report

The authors have addressed all comments. I have no further comments.